A decrease in reports on road-killed animals based on citizen science during COVID-19 lockdown

Dörler Daniel daniel.doerler@boku.ac.at
http://orcid.org/0000-0002-0083-4908 Heigl Florian
Institute of Zoology, University of Natural Resources and Life Sciences , Vienna , Austria
Yoccoz Nigel
Electronic publication date: 2021 Nov 23
Publication date: 2021
Volume: 9
Electronic Location ID: e12464
Received 2021 May 7; Accepted 2021 Oct 19
Copyright: © 2021 Dörler and Heigl
Copyright year: 2021
Copyright holder: Dörler and Heigl
License: This is an open access article distributed under the terms of the Creative Commons Attribution License, which permits unrestricted use, distribution, reproduction and adaptation in any medium and for any purpose provided that it is properly attributed. For attribution, the original author(s), title, publication source (PeerJ) and either DOI or URL of the article must be cited.
License URL: https://creativecommons.org/licenses/by/4.0/

Keywords: Road ecology, Citizen science, Roadkill, COVID-19, Lockdown, Travel behavior

Funding: BOKU Vienna Open Access Publishing Fund Open access funding provided by BOKU Vienna (Open Access Publishing Fund). The funders had no role in study design, data collection and analysis, decision to publish, or preparation of the manuscript.

==============================
Background

To avoid the uncontrolled spread of COVID-19 in early 2020, many countries have implemented strict lockdown measures for several weeks. In Austria, the lockdown in early spring has led to a significant drop in human outdoor activities, especially in road traffic. In Project Roadkill, a citizen science project which aims to collect data on road-killed animals, we observed a significant decrease in reported roadkills.

Methods

By asking the citizen scientists through a survey how their travelling routines were affected, we investigated if the observed decrease in roadkills was grounded in less animals being killed by traffic, or in citizen scientists staying at home and thus reporting less road-killed animals.

Results

A majority of the respondents stated that they felt to have reported less roadkills during the lockdown, regardless if they changed their travelling routine or not. This observation in combination with the overall decrease in road traffic indicates that fewer animals were killed during the lockdown. We conclude that when analyzing citizen science data, the effects of lockdown measures on reporting behaviour should be considered, because they can significantly affect data and interpretation of these data.

Introduction

The global COVID-19-pandemic is affecting human activities and lives in a way like never before in our lifetime. To avoid unnecessary deaths of thousands of people due to excessive demands on our health systems (Anderson et al., 2020), most governments all over the planet have installed strict measures to slow down infection rates in the population (WHO, 2020). In Europe, many people have experienced curfews or strict limitations to leaving the house or apartment since March 2020 (Hale et al., 2020). Activities have been reduced to necessary services, such as the transportation and selling of food or drugs (Hale et al., 2020). Austria has been among the first countries in Europe to implement strict limitations to leaving one’s home in mid-March 2020, more exactly from March 16 to April 14 (Hale et al., 2020). During this time period people living in Austria only were allowed to go outside for five reasons: (i) going to work if one is working in an essential job (e.g., in supermarkets or pharmacies), (ii) to go shopping for food or drugs, (iii) to help other people who cannot care for themselves, (iv) to take a short walk outside always keeping at least one meter distance to other people and (v) to save your life in case of life-threatening events (e.g., fire) in your home (Republik Österreich, 2020). This meant that many people were working from home, were working short hours or lost their jobs, reducing activities or travelling in the country to a minimum (Poledna et al., 2020). This reduction of human activities in Austria, but also in several other countries, has resulted in ecological effects, such as a better air quality due to a dramatic decrease in industrial activities and traffic (Zambrano-Monserrate, Ruano & Sanchez-Alcalde, 2020). Furthermore, many reports on “nature recovering” during lockdowns all over the world where shared on popular media and social media, showing wildlife in settled areas (Rutz et al., 2020; Bar, 2020), suggesting that negative effects of factors usually influencing animal activity or habitat suitability (e.g., habitat reduction, noise-, light-, air-, or water-pollution, hunting, or roads) were decreasing in their extent. Helm (2020) states that at the moment the evidence for immediate impacts of the COVID-19 restrictions on wildlife and environmental protection including the impact of reduced road traffic on roadkill numbers is largely anecdotal so far and based on the expected consequences rather than new data.

Roads can have positive or negative influence on animals, depending on the species, season or road characteristics (Rytwinski & Fahrig, 2015). The most direct negative influence of roads on vertebrates is certainly roadkills (Clevenger, Chruszcz & Gunson, 2003).

Several projects focus on collecting data of road-killed animals and study its impact on animal population (Schwartz, Shilling & Perkins, 2020). The aims of these projects are to get an overview of where animals are being road-killed in order to investigate the underlying reasons and to be able to mitigate hotspots. Projects in the USA, Taiwan, South Africa, Great Britain or the Czech Republic involve citizen scientists in data collection (see http://globalroadkill.net/). In citizen science, research projects are conducted in collaboration with or completely by laypeople (Vohland et al., 2021). In some projects (e.g., Dieren onder de wielen by Natuurpunt in Belgium; https://old.waarnemingen.be/vs/start) citizen scientists monitor single roads at regular intervals, but in most projects citizen scientists report opportunistic data on road-killed animals during their daily routine (Bíl et al., 2020; Shilling et al., 2020), which is a common approach in many ecological citizen science projects (Van Strien, Van Swaay & Termaat, 2013; Horns, Adler & Şekercioğlu, 2018). In these projects, interested people are reporting data mostly via an app or an online form when they observe a dead animal on the road. Usually, they take a picture, identify the animal species and provide the GPS coordinates of the roadkill (Shilling et al., 2020). Although it is usually more difficult to analyze opportunistic data (Planillo et al., 2021; Van Eupen et al., 2021), most projects have opted for this method, as the involvement of citizen scientists in opportunistic data collection offers many advantages. Through citizen science it is possible to sample a large geographical area (Theobald et al., 2015), to bring new expertise into the project, and last but not least, provide opportunities for topical education and science communication (e.g., Shilling, Perkins & Collinson, 2015; Vercayie & Herremans, 2015; Heigl et al., 2017; Kelemen-Finan, Scheuch & Winter, 2018; Chyn et al., 2019). In general, citizen science is already massively contributing to international biodiversity monitoring (Chandler et al., 2017).

Due to involvement of the public in the data collection process, we expected an influence of the COVID-19-lockdown measures on reported numbers of road-killed animals. In the US, a decline in wildlife vehicle collisions in California, Idaho and Maine following the stay-at home orders from the government was reported (Shilling et al., 2021). In Australia, an investigation reported a decrease of roadkill numbers by 48% during the lockdown in April 2020 on a road section consisting of 18 km, which was studied on a regular basis for the last 5 years (Driessen, 2021). Łopucki et al. (2021) observed a decrease in road-killed hedgehogs by 50% in comparison to previous 2 years of monitoring in the city of Chelm (Poland) during lockdown. Furthermore, a recent study showed a decline in roadkill numbers from 11, mostly European, countries (Bíl et al., 2021), not including Austria. However, after the lockdown the numbers rose again to the previous level.

These investigations all have in common that they are based on data that could also be consistently collected during lockdown periods (e.g., police reports, professional monitoring, carcass removal), but not on citizen science data. One study even excluded data based on citizen science from analyses due to potential confounding effects during lockdown periods (Bíl et al., 2021). It is therefore crucial to understand what kind of effects lockdowns have on citizen science data to address potential biases or confounding effects.

In Austria, Project Roadkill is conducted by a team of four people since 2013. In the project, we aim to minimize the number of roadkills. For this, we investigate the influencing factors of traffic and of the landscape surrounding roads on vertebrates. The project is in its first stage, which is to get an overview of the numbers and distribution of roadkills using a citizen science approach (Heigl et al., 2017; Bíl et al., 2020). In Project Roadkill citizen scientists report road-killed animals they encounter on roads via apps for Android or iOS, or by the project’s website (www.roadkill.at/en).

The project allows for every vertebrate species to be reported, including small mammals like hedgehogs or mice, amphibians, reptiles and birds, but also wildlife such as deer, wild boar or hares, that are categorized as huntable wildlife in Austria (Sternath & Dutter, 2006). Participants mainly report from suburban areas and high level roads (Heigl et al., 2016). The citizen science-approach has been shown to be a feasible one to investigate roadkill (Shilling, Perkins & Collinson, 2015; Heigl et al., 2017; Chyn et al., 2019; Yue, Bonebrake & Gibson, 2019; Bíl et al., 2020; Englefield et al., 2020), since citizens cover long stretches of roads every day (e.g., when riding their bikes as a hobby or when commuting to work). Therefore, during the COVID-19 lockdown we immediately experienced a decrease in the number of roadkill reports compared to the previous years (Fig. 1).

Figure 1 Comparison of the mean number of roadkills from 2016 to 2019 (black dashed line) with the number of reported roadkills from 2020 (red line).

Strict lockdown measures were effective in calendar weeks 12 to 15 (indicated by the red highlighted area), with gradual relief in the following weeks. Error bars are given for the years 2016–2019. (A) Overall number of reported roadkills. (B) Number of reported mammal roadkills. (C) Number of reported amphibian roadkills. (D) Number of reported hedgehog (Erinacaeus sp.) roadkills. (E) Number of reported Common toad (Bufo bufo) roadkills.

In previous years, numbers of reported roadkills in many species were rising in spring due to animal migration to breeding sites (e.g., Common toad, Bufo bufo) or for search of food (e.g., hedgehogs, Erinacaeus sp.) after hibernation. The overall rise in numbers of roadkill reports was less pronounced in spring 2020 than in the spring seasons of the previous years. However, at this time we did not know if this drop was a result of fewer animals being killed on roads due to reduced traffic as was the case in previous investigations (Nguyen et al., 2020; Driessen, 2021). Another explanation could be that the participants in the project were traveling less on roads and therefore could not report the same amount of roadkills than before, although the number of roadkills would be unchanged. The difficulty of distinguishing between ecological and sociological factors (observer bias) influencing the data collected in ecological citizen science projects is well known and has been investigated on several occasions. For example, we know that weekend bias occurs in some projects (Courter et al., 2013; Cooper, 2014) or that data is more often reported in the vicinity of settlements or roads (Johnston et al., 2020). However, the behavioral changes brought about by the lockdown measures are unique and need to be studied in detail to see how these changes affect the ecological data collected in citizen science projects. Still, these two influences must be strictly differentiated in the interpretation of the data.

Therefore, the aim of the present study is to investigate how the strict limitations on leaving ones home in Austria during the first COVID-19 lockdown in spring 2020 was affecting the data collection in Project Roadkill. We tested the hypothesis that the drop of roadkill reports was due to changed travel behavior of our participants by conducting a survey among Austrian project participants. The results of this investigation will have implications for further analyses regarding data collected during the lockdown period, not only in Project Roadkill but in all citizen science projects which rely on opportunistic ecological data collection.

Materials & methods

To test the hypothesis, we conducted a survey among participants of Project Roadkill. At the beginning of April 2020, when this survey was sent out, Project Roadkill had 853 citizen scientists coming from 33 countries. The survey was designed for participating citizen scientists in Austria because (I) the focus of the project is on Austria, (II) most data is submitted by Austrian citizen scientists, and (III) the specific lockdown measures differed between countries.

The survey was open from April 20–May 04, 2020 and consisted of seven primary questions and nine sub-questions that asked for more detailed information if specific primary questions were stated positively. The survey language was German. A translation of the survey is available in Appendix 1. Therefore, the participants had to answer 7 questions minimum and 16 questions maximum.

Citizen scientists were informed about the survey via the in-app push message function on April 20, 2020. Citizen scientists who opened the app got a notification on the main screen that a new message has arrived. By clicking on this notification, they could read the message including a link to the survey. The push message was sent to all citizen scientists regardless of the area they are reporting roadkills from. Therefore, the message specifically asked for citizen scientists in Austria. Citizen scientists who report roadkills using the website did not get the push-message. Additionally, we sent the survey link via the newsletter mailing list of Project Roadkill on April 20, 2020. This information was also given in German and was again specifically asking citizen scientists in Austria to participate in the survey. Only people actively signing up for the newsletter receive these newsletters. Consequently, the newsletter recipients do not cover all citizen scientists of Project Roadkill and might also include people who are not reporting roadkills at all but are just interested in the topic. Reminders for the survey were sent after 3 days and 10 days, respectively, via push messages and the newsletter.

The survey was conducted with the software Lime Survey (version 3.21.1). In the following table (Table 1) we list all questions types we used for the individual questions.

Table 1 Detailed list of all answers to the primary questions of the survey.

Question	Response options	I felt I reported the same amount of roadkills	I felt I reported more roadkills	I felt I reported less roadkills	Responses in total per response option	
Has the length of your ways changed?	Increased	3 (4%)	1 (1%)	1 (1%)	5 (6%)	
Reduced	8 (10%)	0 (0%)	29 (38%)	37 (48%)	
No change	14 (18%)	2 (3%)	19 (25%)	35 (45%)	
Has the frequency with which you travel on roads changed?	Increased	2 (3%)	3 (4%)	1 (1%)	6 (8%)	
Reduced	12 (16%)	0 (0%)	35 (45%)	47 (61%)	
No change	11 (14%)	0 (0%)	13 (17%)	24 (31%)	
Has the route of your ways changed?	Yes	5 (6%)	0 (0%)	14 (18%)	19 (25%)	
No	20 (26%)	3 (4%)	35 (45%)	58 (75%)	
Has the type of roads you use changed?	Yes	9 (12%)	0 (0%)	9 (12%)	18 (23%)	
No	16 (21%)	3 (4%)	40 (52%)	59 (77%)	
Has the type of transportation you use changed?	Yes	11 (14%)	0 (0%)	14 (18%)	25 (32%)	
No	14 (18%)	3 (4%)	35 (45%)	52 (68%)	
Note:

Absolute numbers of answers to each question are given. In addition, the percentage in relation to the total number of all submitted answers (n = 77) is given in brackets.

Since the exceptions of the curfew rules during the lockdown were unclear to many in this first phase of the pandemic, we ensured that participation in the survey was completely anonymous to avoid any potential concerns by citizen scientists to face legal consequences if they still had to travel during that time period.

So-called primary questions are questions that ask for a general trend (e.g., Did the number of roadkills you reported change in the period from 16.3.2020 to 13.4.2020?), could be stated with yes or no in most cases (exception was question F2, where participants could decide between three different answers, see Table 2 and Appendix), and have only one digit in the question code (e.g., F1, F2, F3). Secondary questions asked for more details and usually had a “list (option field)” or a “dual matrix”-structure. Some questions were skipped if answers to primary questions indicated that no more details are necessary (e.g., F3: Question 3: Has the length of your routes from which you potentially report roadkills changed in the period from 16.3.2020 to 13.4.2020? If the answer was “No”, secondary questions F31 to F312 were skipped).

Table 2 List of question types used for the survey.

Question code	Type	
F1	Yes/No	
F2	List (option fields)	
F3	Yes/No	
F31	List (option fields)	
F311	List (option fields)	
F312	List (option fields)	
F4	Yes/No	
F41	List (option fields)	
F411	List (option fields)	
F412	List (option fields)	
F5	Yes/No	
F51n	Dual matrix	
F6	Yes/No	
F61n	Dual matrix	
F7	Yes/No	
F71	Dual matrix	
Note:

The question code corresponds to the individual questions in the survey. The different question types were extracted from Lime Survey.

The descriptive data analyses were conducted in Microsoft Excel (version 2002, build 12527.21104) using Pivot Charts. Additionally, we conducted Chi-square tests to analyze potential influences of changes (i) in daily route length, (ii) frequency of moving on roads, (iii) overall routing (i.e., if the respondents changed areas through which they move regularly), (iv) road types and (v) modes of transportation using R Studio (V 1.4.1103 (R Core Team, 2018)) with the package gmodels V 2.18.1 (Warnes, Bolker & Lumley, 2018).

Results

In total, 77 persons completed the survey. In comparison, 179 persons actively reported roadkills in Austria 12 months before the start of the survey. Almost two thirds (64%) have indicated that they estimated to have reported less roadkills than before the lockdown, 32% said to have reported the same amount of roadkills, and only 4% felt to have reported more roadkills than before. A detailed list of the answers can be seen in Table 2.

When asked if anything has changed in the way respondents were moving on roads, more than two thirds (69%) stated that the frequency with which they were moving on roads has changed, opposed to 31% who stated they were moving with the same frequency as before. In addition, more than half of the respondents (55%) stated that the length of their ways has changed, whereas 45% stated that the length of their ways did not change at all. The routes, however, did not change for most respondents (75%), as did the type of roads used (77%); 23% said they used other roads than before. The type of transportation used for travelling on roads also did not change for most respondents (68%); 32% said they changed the type of transportation used in the time during the lockdown.

To test our hypothesis that the drop of roadkill reports was due to a changed travel behavior of our participants, we compared the answers from the respondents which stated that they did not change the frequency of travel, length of route, the type of road or the type of transportation used to those who reported changes in their travel behavior due to the lockdown. Here we found no significant differences between the two groups (p = 0.454). When analyzing all answers from respondents who stated that they did not change the length of their ways, the frequency with which they travelled their route, the type of the road and the type of transportation used, we found a majority of 60% of respondents who felt to have reported less roadkills compared to 40% who felt to have reported the same amount of roadkills.

In Table 2, we see that most respondents stated that they reduced the length of their ways and the frequency with which they travel. A total of 55% of the respondents who reduced the length of their ways and who felt to have reported less roadkills reported a reduction in route length by 75–100%, compared to 50% of respondents who felt to have reported the same amount of roadkills. Furthermore, 57% of those respondents who stated that they reduced the frequency with which they travel on roads and who felt to have reported less roadkills also said that they reduced the frequency between 75% and 100%, compared to 25% of respondents who felt to have reported the same amount of roadkills. Chi-square tests also revealed that the factors reduction of length and reduction in frequency were also the only two that significantly influenced the number of roadkills participants felt to have reported (p = 0.02 for reduction of length and p = 2.92e−08 for reduction in frequency; change in routes: p = 0.43; change in roadtype: p = 0.15; change in transportation: p = 0.19). People who said that the length of their routes has been reduced estimated to report predominantly less roadkills (81%) or the same amount of roadkills (19%). Respondents who estimated to have reported more roadkills than before predominantly indicated that they travelled either longer routes or the length of their routes did not change at all (Fig. 2). However, the interpretation of the answers must be made carefully since only three respondents stated that they reported more roadkills.

Figure 2 Mosaic plot of the estimated change in the reported number of roadkills by respondents compared to the length of their routes.

Colours indicate changes in estimated number of reported roadkills (red = increase, green = no change, blue = reduction).

When we asked for change in frequencies with which people travel on their routes, we saw similar results (Fig. 3). Respondents who reported that their frequency of moving on roads increased estimated to have reported more roadkills compared to people who stated that they reported less or the same amount of roadkills, which were moving predominantly less frequently on roads than before.

Figure 3 Mosaic plot of the estimated change in the reported number of roadkills by respondents compared to the frequency with which they were moving on roads.

Colours indicate changes in estimated number of reported roadkills (red = increase, green = no change, blue = reduction).

The answers indicate that the routes, types of roads used and types of transportation used did not change for a majority of the respondents. All three questions concerning these potential lockdown effects were answered negatively with at least a two-third majority. Consequently, we also could not detect a significant influence of these aspects on the estimated number of reported roadkills. Furthermore, people who felt to have reported more roadkills than before the lockdown did not experience any change in routes, types of roads used or change in transportation mode.

Discussion

The COVID-19-measures have clearly influenced participants reporting behaviour in Project Roadkill as the feedback from our survey indicates. Almost two thirds of the respondents stated that they estimated to have reported fewer roadkills during the lockdown than before, which is in line with the overall reduced number of reported roadkills during the lockdown period (Fig. 1A). However, in some species (groups) fluctuations are usually very high this time of year as can be seen in Fig. 1. Particularly amphibian roadkill numbers vary significantly this time of year, depending on spring temperatures (e.g., Timm, McGarigal & Compton, 2007; Scott, Pithart & Adamson, 2008; Ficetola & Maiorano, 2016). Therefore, the observed number of amphibian or Common toad roadkills is within the expected fluctuations (Figs. 1C and 1E), although on a very low level. Roadkill numbers of other species that also hibernate but are not as temperature sensitive as amphibians, such as mammals (e.g., Wang, 1989), show smaller fluctuations. Numbers for these species in 2020 (e.g., hedgehogs) are outside the range of expected fluctuations (Figs. 1B and 1D).

Moreover, we could not detect any significant differences regarding the estimated observed roadkills between people who changed their travelling behavior in any way and those respondents who changed nothing at all.

We can see that most respondents experienced profound effects of the lockdown on their travelling routine. The most influential changes according to our results are the reduced frequency with which people travel on roads and the reduction in route length. This change in travelling routine seems to be the dominant reason for reporting less roadkill, which is in line with previous studies which found that monitored road sections had to be inspected several times a week to cover all potential roadkills (Bager & Da Rosa, 2011; Ratton, Secco & da Rosa, 2014). Reasons for this are low persistence rates especially for smaller vertebrates such as amphibians or small mammals (Santos, Carvalho & Mira, 2011; Santos & Ascensão, 2019), as they dissolve after a very short period of time, and scavengers that remove carcasses of road-killed animals of roads (Ratton, Secco & da Rosa, 2014). People who travel longer routes also have a higher chance to encounter road-killed animals. The importance for route length and travel frequency is also confirmed when we look at the group of respondents which felt to have reported more roadkills. Although only a very small number of people (4% of all respondents), they consistently only reported changes in route length and/or travel frequency, but no overall route changes or changes in types of roads or modes of transportation.

Interestingly, however, there was no significant difference in the perceived number of roadkill between the two groups who changed their travel behaviour and those who did not. This result could indicate that possibly fewer animals were road-killed on Austria’s roads during the lockdown. This indication is also supported by the fact that fewer vehicles were on the road during the lockdown and thus the probability of a roadkill was reduced. The official traffic statistics show a significant decrease in the number of vehicles on Austrian motorways in March (45% decrease) and April (54% decrease) compared to the mean number of vehicles on Austrian motorways for January and February 2020 (ASFINAG, 2020).

Several respondents reported a change in the mode of transportation from car to bicycle or going on foot. Although traveling shorter routes, these respondents would have had the chance to discover roadkills they could not see when driving a car e.g., smaller amphibians or mammals. These animals are often overlooked when driving a car due to high velocity (Slater, 2002; Erritzoe, Mazgajski & Rejt, 2003; Guinard, Prodon & Barbraud, 2015). Nonetheless, a majority of the respondents who changed from car to going on foot felt to report less roadkills than before. One explanation for this could be that these respondents previously moved in agricultural or silvicultural areas and were moving in settled areas only during the lockdown, where at least some smaller vertebrate species are killed less often (Rodríguez-Castro et al., 2017). However, our survey results for this particular group of respondents do not confirm this explanation. The respondents were moving in settled and agricultural areas in equal shares during the lockdown.

Conclusions

COVID-19 has clearly affected Project Roadkill and its participants. We experienced a significant drop in the number of reported roadkills in 2020 during the lockdown weeks compared to the mean number of reported roadkills in the years 2016–2019. In previous years there have been two peaks in spring during which many road-killed animals have been reported. In 2020, the first peak has been less pronounced and the second peak is missing completely. Two explanations for this are possible, the second being more likely in our view. Firstly, through the lockdown measures data especially from those animals that get active after hibernation in early spring, such as many amphibians or mammals (e.g., hedgehogs), was not reported due to reduced travel behaviour of our citizen scientists. Secondly, our results indicate that there could be an overall decrease in roadkills due to reduced vehicle numbers on roads during the lockdown. Additionally, travel frequency and travel length seem to be the main factors for a decrease in roadkill reports. We can also see that the majority of the respondents have encountered less roadkills than usual. The investigation shows the duality of Project Roadkill. If fewer participants in our project are on the roads and if this is representative of society, the reports in the project are reduced on the one hand, but also the negative effects of road traffic on vertebrates. Our study indicates that future data analysis based on citizen science projects should take into account the potentially changed reporting behavior of citizen scientists during the COVID pandemic in order to avoid incorrect ecological conclusions.

Supplemental Information

Supplemental Information 1 Questionnaire (in original German).

Click here for additional data file.

Supplemental Information 2 Answers to questionnaire (in original German).

Click here for additional data file.

Supplemental Information 3 Questionnaire (translated to English).

Click here for additional data file.

Supplemental Information 4 Answers to questionnaire (translated to English).

Click here for additional data file.

We would like to thank all participants of Project Roadkill, who dedicate their time to the collection of data on road-killed animals all over the world and especially those participants in Austria, that answered the survey and provided invaluable insight on how COVID-19 is affecting their lives. Very special thanks go out to all people who are carrying us all through this pandemic, among them doctors, nurses, employees of supermarkets and many more.

Additional Information and Declarations

Competing Interests

Author Contributions

Data Availability

The authors declare that they have no competing interests.

Daniel Dörler conceived and designed the experiments, performed the experiments, analyzed the data, prepared figures and/or tables, authored or reviewed drafts of the paper, and approved the final draft.

Florian Heigl conceived and designed the experiments, performed the experiments, analyzed the data, prepared figures and/or tables, authored or reviewed drafts of the paper, and approved the final draft.

The following information was supplied regarding data availability:

The raw data from the survey and the questions are available in The Supplementary Files.

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
