# Peer review of "A decrease in reports on road-killed animals based on citizen science during COVID-19 lockdown"

_PeerJ, doi:10.7717/peerj.12464_

## Round 0.1 · original submission · Major Revisions

As the reviewers point out, this is an interesting study exploring potential confounding effects of reporting rates on estimates of road-killed animals. The reviewer made constructive comments that will significantly improve the manuscript.

Given that the survey of citizen scientists resulted in a relatively small sample (not a criticism!), you do not need to report percentages with two decimals (ie 63.63% for example is rather meaningless when the uncertainty around 63% is likely to be ca 10%: 63% is good enough). Also, the small sample size leads to low precision or if you prefer low power, so the absence of significant differences does not mean that such differences may not matter.

·

Basic reporting

Language should be checked, particularly in some points that I highlight in specific comments, where the clarity should be enhanced.

Figure 1: In order to show the variation across years, I would plot average line + standard deviation bars, or authors can plot one black line for each year before 2020. This way it will be easier to detect differences in the number of reports between 2020 and previous years.

Figures 2 and 3: the font is really too small, it is impossible to read without zooming in. There is a typo in the bottom label “resondents” in both figures. Columns should be ordered less/same/more roadkills or more/same/less, to make it the understanding of the plots more intuitive.

What is the meaning of the last column of Table 1?

Experimental design

no comment

Validity of the findings

no comment

Additional comments

General comment:

The authors present an interesting work on the effect of lockdown on the number of reports of roadkilled animals. Generally, I think that the topic is interesting, however I have some concerns. My main concern is the language and the presentation of figures. Language should be checked, particularly in some points that I highlight in specific comments, where the clarity should be enhanced.

Figure 1: In order to show the variation across years, I would plot average line + standard deviation bars, or authors can plot one black line for each year before 2020. This way it will be easier to detect differences in the number of reports between 2020 and previous years.

Figures 2 and 3: the font is really too small, it is impossible to read without zooming in. There is a typo in the bottom label “resondents” in both figures. Columns should be ordered less/same/more roadkills or more/same/less, to make it the understanding of the plots more intuitive.

What is the meaning of the last column of Table 1?

My last concern is about discussion. It is a bit confusing, and I would rearrange it in order to avoid repetitions of result. The structure should reflect the main results: i) drop in roadkill reports ii) people moving less iii) people felt to report less roadkills even if their travelling did not change.

Specific comments:

Lines 50-52: The sentence is not clear, I would rephrase that. For instance, “roads” can’t “decrease in extent”: eventually the negative effect of roads can decrease. Additionally, the listed factors do not influence “behavior or habitats” but “activity and habitat suitability” might be preferable as terms. Rephrasing the sentence will help.

L52: Start the sentence with “Roads can have positive or negative influence on animals, depending on…”.

L66-67: Are all the project participant also writing this manuscript? Otherwise, the sentence should be rephrased because it seems that there are just two people in roadkill project (I see at least 4 on the website, so I suggest to rephrase the sentence to avoid misunderstandings).

L69: Rephrase this part of the sentence, it is not clear.

L 69: Sentence starting at the end of this line should start with “The project is its first stage, etc…”.

L79: change “amphibia” to “amphibians”

L79: “huntable” is not a good category. What is “huntable” in Austria may be not in other countries.

L81: “amphibians”

L81-83: I did not understand the sentence. There are many example of usefulness of citizen-science with many groups, not only amphibians. Additionally, why “citizens potentially also cover”? It seems that only the ones that report amphibians travel long distances, while it is not the case for the citizens reporting reptiles/mammals/etc… rephrase the sentence.

L 90-91: Merge the two sentences. In the second one the authors make an example with common toads even if second sentence is about non-amphibians.

L103-105: Too many “however”

L108: delete “your”

L122: “Mai”

L124-125: “You can find a translation of the survey in the appendix (Appendix 1).” -> “A translation of the survey is available in Appendix 1.”

L151-157: What is “F”? I see “Question 1, 1.1, etc” in appendix, no “F”.

L166: Since you shared the survey also with people not participating in the program, it is wrong to say that 77 is 43% of users (you may have also non-users as you state at line 138).

L170: “table” -> “Table” ?

·

Basic reporting

Authors meet all basic reporting requirements of journal - see general comments for author for full review.

Experimental design

Authors meet all experimental requirements of journal - see general comments for author for full review.

Validity of the findings

Authors meet all validity requirements of journal - see general comments for author for full review.

Additional comments

This study describes a decrease in reported roadkill observations in the Project Roadkill database during a COVID-19 lockdown in Austria and examines the underlying cause. The authors explore whether the decrease in reports was due to an actual decrease in roadkilled animals or due to fewer reports from volunteers in Project Roadkill. This is a novel and important study investigating the cause of a decrease of wildlife roadkill observations during COVID-19 lockdown. A few other papers have examined pre- and during-lockdown roadkill observations but have not explored the possibility that fewer roadkill reports were due to fewer volunteers collecting data. This is an interesting and novel topic and provides basic investigation on causes of fewer roadkill reports in Austria, however, the study could use deeper inspection on the following:

- Though not the central question of the study, it would be interesting to see basic quantifications of changes between pre- and during lockdown observations by major taxon groups in Austria. These comparisons could reveal if there were differences in observations of different taxon groups in the two time periods. It could be especially interesting if there were differences in reported roadkill since authors found that many volunteers began walking/biking rather than driving.
- Authors should consider the more inclusive term “community scientist” or volunteer rather than “citizen scientist.” It has been widely accepted and used by many large voluntary science programs such as the U.S. Audubon: “As part of Audubon’s commitment to equity, diversity, and inclusion, we have transitioned from using the term “citizen science” to the more inclusive term “community science.” No matter where a volunteer was born, or how they came to the United States, we value their contribution to our science and conservation programs.” (https://debspark.audubon.org/news/why-were-changing-citizen-science-community-science#:~:text=As%20part%20of%20Audubon's%20commitment,to%20our%20science%20and%20conservation)
- A more in-depth introduction to volunteer-based and opportunistic roadkill observation programs would be helpful so that those unfamiliar will have a better understanding of shortcomings and benefits – expand on lines 56-59 (e.g. Waetjen and Schilling 2017, Chyn et al. 2019, Pagany 2020, and many more). There are also a few other studies on the effect of COVID-19 lockdowns on roadkill numbers not described or cited (e.g. Shilling et al. 2021, Lopuki et al. 2021).
- There is inconsistent usage of road-killed vs roadkilled. Both are valid but authors should choose one consistent spelling.
- Awkward phrasing throughout (a few examples noted below in specific comments) and some sentences need restructuring – could use a revision focused on clarity and flow.
- Figures 2-3: helpful visualizations, but since the frequency of daily routes is “reduced,” “no change,” “increased”, I think it would be more consistent (and smoother) if the number of reported roadkill is in a similar order of “less roadkills,” “same,” then “more roadkills.”

Specific comments by line:

42: “life” vs “live”
60: “wildlife” (no cap)
66-67: Transition to last sentence is awkward and could be smoother.
68: Don’t need “as much as possible”
73: “In Project Roadkill?”
80: “suburban”
90-93: Restructure sentences for clarity. Grammar is a little unclear.
94-98: Restructure for clarity – run on sentence.
107-109: Verbose/wordy – revise for clarity.
110: Delete “a”
124: “May”
166: Cite ‘gmodels’ developers
202: “Chi-square”
215: “We got a similar picture when we asked for” – a bit colloquial. Revise with more technical language.
234: “fewer” vs “less”
300: “roadkills”

---

## Round 0.2 · Minor Revisions

The revised paper is signficantly improved as the reviewer made clear. There are, however, some issues that need to be checked, in particular some figures which seem to be inconsistent with the text.

·

Basic reporting

Authors meet all basic reporting requirements of journal - see general comments for author for full review.

Experimental design

Authors meet all experimental requirements of journal - see general comments for author for full review.

Validity of the findings

Authors meet all validity requirements of journal - see general comments for author for full review.

Additional comments

General comments:

The authors have made major revisions to the manuscript in the introduction and added analyses and figures (though see comment on figures below). Overall, it is much improved and an interesting study.

I believe the authors are referring to the term “community-based science” rather than “community science.” It is true that the term “community-based science” usually refers to efforts led by the wider community and would not be fitting for a university-led project, but “community science” is used in lieu of “citizen science” as a more inclusive term for researcher/university/institution led volunteer data collection projects. This is not a major concern for the paper as it is on inclusive language rather than content, and I respect if the authors decide to continue to use “citizen science” instead as it is widely used. I very recently switched to using “community science” as a term myself.

After the paragraph describing other COVID roadkill studies in lines 86-96, there is an opportunity to discuss how some of these studies may not take possible bias of staying home vs actual decrease into consideration. This would be a good transition to describing project Roadkill and how this particular study is novel.

I appreciate the addition of taxon roadkill figures in Figure 1. Figures 2 and 3 seem to be very different than the figures from the original submission. If I’m reading the mosaic plot correctly, it looks like the proportions of more, same, and less reported roadkills are switched (not just the order). Please double check that this is the correct figure. Currently the figures read as most people reported more roadkills with reduced length and frequency of driving which seems incorrect compared to the results.

Check comma usage throughout. It is unnecessary in many places and missed in others. Some instances noted below.

Additionally, inconsistent usage of “Project/project Roadkill” – sometimes project is capitalized while other times it is not.

Specific comments:

59: comma unnecessary

70: “In some roadkill monitoring projects,” – sentence before is general so specifying is necessary

75: comma unnecessary

76: “provide the GPS coordinates of the roadkill” vs “locate the location of…”

80-82: comma usage and phrasing needs to be revised – “Through citizen science it is possible to sample a large geographical area (Theobald et al., 2015), to bring new expertise into the project, and last but not least, provide opportunities for topical education and science communication”

86: delete “this”

86: “the data collection process,”

87: “In the US,”

111: in () “e.g. when riding their bikes as a hobby or when commuting to work.”

117: “In previous years,”

117-119: revise sentence phrasing and add scientific names

121: “fewer” vs “less”

142: roadkill does not need to be capitalized unless referring to “project Roadkill”

219: “In Table 2,”

261: “Particularly,” vs “Especially”

---

## Round 0.3 · accepted · Accept

You have answered all comments made by the reviewer, and changed figures 2 and 3 accordingly. Note, however, that the legend of the figures is very hard to read (too small) and you will have to provide better figure legends for the final publication.